# Integrated Metabolomics and Lipidomics Analysis Reveals the Mechanism Behind the Action of Chiglitazar on the Protection Against Sepsis-Induced Acute Lung Injury

**DOI:** 10.3390/metabo15050290

**Published:** 2025-04-25

**Authors:** Liu-Liu Lu, Yu-Li Cao, Zhen-Chen Lu, Han Wu, Shan-Song Hu, Bing-Qing Ye, Jin-Zhi He, Lei Di, Xu-Lin Chen, Zhi-Cheng Liu

**Affiliations:** 1School of Pharmaceutical Sciences, Department of Anesthesiology, The Third Affiliated Hospital of Anhui Medical University, Anhui Medical University, Hefei 230032, China; luliuliu810@gmail.com (L.-L.L.); luzhenchen1028@gmail.com (Z.-C.L.); wuuu91627@gmail.com (H.W.); shansonghu183@gmail.com (S.-S.H.); yebingqing1231@gmail.com (B.-Q.Y.); hejinzhi126@gmail.com (J.-Z.H.); dilei@ahmu.edu.cn (L.D.); 2The First Department of Critical Care Medicine of the Second Affiliated Hospital of Anhui Medical University, Hefei 230601, China; caoyuli0901@gmail.com; 3Department of Burns, The First Affiliated Hospital of Anhui Medical University, Hefei 230022, China

**Keywords:** sepsis-induced acute lung injury, metabolic modulation, metabolic disorder, multi-omics, chiglitazar

## Abstract

**Background:** Sepsis-induced acute lung injury (SALI) is a critical clinical challenge with high mortality. Metabolic dysregulation drives SALI pathogenesis, disrupting lung function and energy metabolism. Despite proven benefits, metabolic restoration is underused in sepsis. This study explores chiglitazar’s role in balancing metabolism to protect against SALI. **Methods:** The protective effects of chiglitazar in CLP rats were demonstrated by the survival curve, histological analysis, and immunohistochemical analysis in the lung tissue. Metabolomic and lipidomic analyses of lung tissue samples using gas chromatography–mass spectrometry (GC-MS) and liquid chromatography–mass spectrometry (LC-MS) were performed to evaluate metabolic shifts induced by CLP surgery and chiglitazar pretreatment. The mRNA and protein levels of the underlying targets directing nicotinamide adenine dinucleotide (NAD+) and triglyceride synthesis were analyzed by qPCR and Western blotting. To validate the mechanism by which chiglitazar protected against SALI, the SIRT1 inhibitor EX-527 was applied to human normal lung epithelial (BEAS-2B) cells and another batch of rats to observe its reverse effect against chiglitazar’s action. **Results:** Chiglitazar pretreatment significantly restored NAD+ and improved dysregulated lipid metabolism by enhancing the synthesis of triglycerides (TGs) and suppressing accumulated fatty acids (FAs). The metabolic modulation mediated by chiglitazar was associated with the upregulations of the SIRT1/PGC-1α/PPARα/GPAT3 axis. Co-treatment with EX-527 in LPS-stimulated BEAS-2B cells and CLP rats inhibited the effects of chiglitazar on the aforementioned signaling pathways and worsened the protective effects of chiglitazar on lung injury, respectively. **Conclusions:** Chiglitazar alleviates SALI by restoring NAD+ and TG synthesis, highlighting the balancing of metabolism as a promising therapeutic strategy in the management of SALI.

## 1. Introduction

Sepsis, characterized by a dysregulated host response to infection that leads to life-threatening organ dysfunction, is a prevalent critical illness associated with multiple organ failure. Sepsis remains a leading cause of mortality worldwide [1]. The lung is among the earliest and most frequently affected organs in sepsis, with acute lung injury (ALI) being a common and early manifestation. Unlike damage to organs with compensatory functions, such as the liver or kidneys, sepsis-associated acute lung injury (SALI) can further develop into acute respiratory distress syndrome (ARDS), which significantly contributes to sepsis-related mortality [2,3]. Current clinical therapies for sepsis include a range of treatments, such as antibiotics, vasoactive drugs, and glucocorticoid anti-inflammatory agents. However, antibiotics can lead to antimicrobial resistance and organ damage [4], whereas glucocorticoid anti-inflammatory drugs are often associated with gastrointestinal bleeding, metabolic disorders, and an increased risk of cardiovascular complications [5]. Despite recent advancements in sepsis therapies, the high mortality rate remains a significant concern, primarily due to the unresolved turnovers in immune responses, as well as organ damage resulting from immune system dysregulation and metabolic disorders [6,7].

Metabolic dysregulations are crucial hallmarks of sepsis pathogenesis, roiling energy production pathways and exacerbating organ dysfunction. For instance, immune cell metabolic reprogramming (particularly in macrophages) and enhanced catabolism represent two cardinal features of sepsis-induced metabolic dysregulation. Such metabolic shifts contribute to mitochondrial dysfunctions [8,9,10]. Concurrently, sepsis-driven lipid metabolic disorders characterized by excessive lipolysis, defective fatty acid β-oxidation, and accumulation of cytotoxic lipid species (e.g., oxidized phospholipids and ceramides) disrupt systemic homeostasis [11,12,13]. Additionally, dysregulations of bioactive lipid mediators are critical in sepsis-induced inflammation. Pro-inflammatory lipids, such as ceramides and oxidized phospholipids, activate NLRP3 inflammasomes and contribute to endothelial damage. In contrast, sphingomyelins counteract inflammation via NF-κB suppression [12,14,15]. Lysophosphatidylcholines (LPCs) exhibit context-dependent roles, with certain subtypes suppressing inflammation while others enhance leukocyte chemotaxis [16,17], induce cellular dysfunction, and trigger excessive inflammatory responses via chemotactic signaling cascades [18]. Additionally, the depletion of specialized pro-resolving mediators (SPMs) such as resolvins and protectins impairs the resolution of inflammation, perpetuating tissue damage [19]. These metabolic disturbances synergistically potentiate systemic inflammatory cascades and tissue damage [20,21]. Therefore, restoring energy metabolism balance has become a promising therapeutic strategy to improve SALI.

Increasing numbers of studies have explored the potential of modulating metabolic processes to improve the prognosis of sepsis patients [22,23]. Various glucose-lowering agents have been used to treat sepsis. For instance, insulin is revealed to suppresses pro-inflammatory cytokines like TNF-α and IL-6 while enhancing endothelial function and improving short-term survival in septic patients [24,25]. Similarly, metformin is used to reduce sepsis mortality through AMPK-dependent inhibition of the nuclear factor kappa-B (NF-κB) signaling, yet its clinical utility is hampered by lactic acidosis in critically ill populations [26]. Empagliflozin, an SGLT2 inhibitor, exemplifies this paradigm by attenuating sepsis-induced acute kidney injury via NLRP3 inflammasome suppression, though its renoprotective effects are secondary to immunomodulation rather than metabolic or structural repair [27]. Likewise, thiazolidinediones (TZDs) such as rosiglitazone and pioglitazone, as peroxisome proliferator-activated receptor γ (PPARγ) agonists, were shown to improve septic outcomes primarily through suppressing NLRP3 inflammasome activation and cytokine storms [28,29]. Similarly, Pawlak et al. revealed that peroxisome proliferator-activated receptor α (PPARα) exerts broad anti-inflammatory effects by suppressing pro-inflammatory signaling pathways, suggesting its potential to improve sepsis outcomes through systemic inflammation modulation [30]. However, these studies predominantly attribute glucose lowering drug-mediated organ protection to the suppression of inflammation rather than metabolic reprogramming-related organ protection. Although sepsis mortality is tightly linked to irreversible organ damage stemming from metabolic collapses, including energy deficiency-mediated mitochondrial dysfunctions and tissue hypoxia, limited studies have investigated the protective effects of metabolic modulators against sepsis-induced organ injury through their actions on the reversal of sepsis-associated metabolic dysregulation.

Chiglitazar, a pan-PPAR agonist, acts as a multitarget regulator for balancing both glucose and lipid metabolism. It is therefore supposed to ameliorate SALI, but the actual effect and corresponding mechanism remain unclear. Metabolomics and lipidomics are essential techniques for studying metabolic changes induced by diseases or pharmaceutical interventions [31]. These technologies provide comprehensive insights into the biochemical changes occurring during sepsis, facilitating a deeper understanding of disease mechanisms and the effects of therapeutic interventions [32,33]. Using integrated metabolomics and lipidomics analysis, as well as further tests on the expressions of relevant targets, this study aims to understand whether and how chiglitazar mediates lung protection from the point of view of its effects on sepsis-associated metabolic disorders.

## 2. Materials and Methods

### 2.1. Drugs and Materials

Chiglitazar was obtained from Chipscreen Biotechnology (Shenzhen, China). Pioglitazone was purchased from Bidepharm (Shanghai, China). Pentobarbital sodium was procured from Halingbio (Shanghai, China). Methanol was acquired from Fisher Scientific, Hampton, NH, USA. Acetonitrile was supplied by J&K Scientific (Beijing, China). Isopropyl alcohol, dichloromethane, pyridine, methoxamine hydrochloride, formic acid, phenylmethylsulfonyl fluoride, and N-Methyl-N-(trimethylsilyl)trifluoroacetamide were obtained from Sigma-Aldrich (Shanghai, China). All the internal standards used in metabolomic and lipidomic analyses were obtained from Sinoptinciple Chemical Technology (Shanghai, China). Paraformaldehyde (4%) was purchased from Solarbio (Beijing, China). Lipopolysaccharide from *E. coli* O55:B5 was purchased from Sigma-Aldrich (Shanghai, China). High-glucose DMEM medium was supplied by BasalMedia (Shanghai, China). Fetal bovine serum was obtained from Royacel (Lanzhou, China). Penicillin-streptomycin solution was procured from Procell (Wuhan, China). EX-527 was sourced from MedChemexpress (Monmouth Junction, NJ, USA; Cat. No. 49843-98-3). The following primary antibodies were utilized: PPARα (Abcam, Cambridge, UK; Cat. No. AB314112); SIRT1 (Cell Signaling Technology, Danvers, MA, USA; Cat. No. 9475); GPAT3 (Proteintech, Wuhan, China; 20603-1-AP); PGC-1α (ZEN-BIO, Chengdu, China; Cat. No. 381615); and β-actin (Affinity, Changzhou, China; Cat. No. AF7018). Horseradish peroxidase (HRP)-conjugated goat anti-rabbit/mouse secondary antibodies were purchased from Elabscience (Wuhan, China; Cat. No. E-AB-1003).

### 2.2. Animal Model

Male Sprague-Dawley rats of SPF grade (220–250 g, sourced from Liaoning Changsheng Biotechnology Co., Ltd., Benxi, China) were housed under standardized environmental conditions. Ethical approval for the animal experiments was obtained from the Animal Experimentation Committee of Anhui Medical University (Approval Number: LLSC20240853).

Five experimental batches were designed to evaluate chiglitazar’s efficacy, safety, and molecular mechanisms in sepsis. The first batch assessed survival rates across Sham (control), CLP (sepsis model), and CLP+Chi groups treated with chiglitazar at 2.5, 5, or 10 mg/kg (denoted as CLP+Chi 2.5, CLP+Chi 5, and CLP+Chi 10, respectively). Subsequent batches expanded the scope to include dose-dependent effects (Batch 2), multi-omics profiling (Batch 3), and SIRT1 inhibitor interactions (Batches 4 and 5). All groups consisted of Sham (underwent laparotomy and suturing only and served as the control group), CLP (underwent cecal ligation and puncture (CLP) surgery, as described previously [34]), and chiglitazar-treated rats (CLP+Chi at specified doses), with additional controls (e.g., pioglitazone (10 mg/kg), EX-527 (5 mg/kg)), as fully detailed in Appendix A. Sample sizes were standardized to *n* = 10 per group, except the Sham (*n* = 6) and CLP groups (*n* = 10–12), to account for expected mortality rates in the sepsis models. The experimental workflow, group assignments, and batch-specific objectives are comprehensively visualized in Figure 1.

Prophylactic administration was conducted in rats for five consecutive days prior to the induction of the disease model. Specifically, rats in the treatment group received chiglitazar via oral gavage daily, while the positive control group was administered pioglitazone with the same method and EX-527 was delivered through intraperitoneal injection [35]. Concurrently, the Sham and CLP groups received an equal volume of double-distilled water following the same administration schedule. After the CLP surgery, rat survival was meticulously monitored and recorded at 6, 12, 24, 36, 48, and 72 h, and a survival curve was subsequently generated to assess the protective efficacy of the interventions.

### 2.3. Samplings

The rats were sacrificed under anesthesia 24 h after modeling procedures, and bronchoalveolar lavage fluid (BALF) and lung tissue were collected. The left lungs were lavaged three times with 0.9% saline using 5 mL syringes. The lavage fluid was centrifuged at 1500 rpm for 10 min and the upper layer of clear fluid was collected as BALF. The BALF and lung tissue samples were subsequently stored at −80 °C until further analysis.

### 2.4. Tissue Weighing and Histochemical Analysis

The wet/dry (W/D) weight ratio was calculated to assess the occurrence of pulmonary edema in rats. The upper lobe of the right lung was excised, and its weight was recorded. The lung tissue was then placed in an incubator at 60 °C for 48 h to determine the dry weight, whereby the W/D ratio was calculated.

For histochemical analysis, lung tissues were fixed with 4% paraformaldehyde, paraffin-embedded, and sectioned at a thickness of 5 μm for H&E staining. A semi-quantitative inflammation scoring system was used based on the following criteria: 0 = normal, 1 = mild edema or neutrophil infiltration, 2 = moderate edema or neutrophil infiltration without significant alveolar space reduction, and 3 = severe inflammatory cell infiltration with significant alveolar space reduction and lung destruction [36]. The assessment was conducted in a blinded manner by two professional researchers.

### 2.5. Immunohistochemical Analysis

Tissue sections were deparaffinized, rehydrated, and treated with 3% H_2_O_2_ for 30 min, followed by incubation with normal goat serum. Primary antibodies targeting zonula occludens-1 (ZO-1) (1:500, ab221547, Abcam) and Occludin (1:200, ab216327, Abcam) were applied overnight at 4 °C. The sections were washed and subsequently incubated with biotinylated secondary antibodies, visualized under a microscope, and analyzed using ImageJ software (version 1.54d).

### 2.6. Cell Culture

Human normal lung epithelial cells (BEAS-2B) were obtained from the Cell Bank of the Chinese Academy of Sciences (Shanghai, China). BEAS-2B cells were cultured in high-glucose DMEM (Dulbecco’s Modified Eagle Medium) medium supplemented with 10% fetal bovine serum (FBS), 100 μg/mL penicillin, and 100 μg/mL streptomycin in a humidified incubator containing 5% CO_2_ at 37  °C. Upon reaching 70–80% confluence, the cells were trypsinized, split at a ratio of 1:3, and further cultured.

### 2.7. Cell Viability Assay

Cell viability was assessed using a Cell Counting Kit-8 (CCK-8) assay (GlpBio Technology, Montclair, CA, USA). BEAS-2B cells were seeded in a 96-well plate and incubated with different concentrations of chiglitazar for 24 h to determine the optimal concentration for subsequent experiments.

### 2.8. Enzyme-Linked Immunosorbent Assay (ELISA)

BEAS-2B cells were stimulated with lipopolysaccharide (LPS) (1 μg/mL) and various concentrations (8, 16, or 32 μM) of chiglitazar, or with chiglitazar 32 μM in combination with EX-527 (5 μM) for 24 h. The corresponding cell culture supernatants were collected by centrifugation. Enzyme-linked immunosorbent assay (ELISA) kits (Beyotime, Shanghai, China) were used to measure the levels of inflammatory cytokines, including interleukin-1β (IL-1β), interleukin-6 (IL-6), and tumor necrosis factor-α (TNF-α), in the cell supernatants and BALF according to the manufacturer’s instructions.

### 2.9. Metabolomics Methodology

Metabolomics analysis was conducted using both gas chromatography–mass spectrometry (GC-MS) and liquid chromatography–mass spectrometry (LC-MS) techniques, as outlined below.

GC-MS-Based Analysis in Lung Tissue: Fifty milligrams of accurately weighed lung tissue was homogenized for 5 min in an ice bath using 500 μL of methanol–water mixture. The mixture was incubated at room temperature for 15 min, vortexed for 30 s, and then centrifuged at 15,000 rpm for 15 min. The supernatant was collected and freeze-dried to remove residual solvents. For the subsequent derivatization reaction, 80 μL of Methoxamine hydrochloride (20 mg/mL in pyridine) was added and reacted at 37 °C for 90 min, followed by the addition of 80 μL of N-Methyl-N-(trimethylsilyl)trifluoroacetamide (MSTFA), and further reacted at 37 °C for 1 h. The samples were then analyzed using GC-MS (QP-2010 system, Shimadzu). A SH-Rxi-5Sil MS column (L30 m, ID 0.25 mm, DF 0.25 μm; Shimadzu) was used for metabolite separation. As for the column temperature program, the initial temperature was 80 °C, which was increased to 180 °C at a rate of 5 °C/min, then to 200 °C at a rate of 2 °C/min, followed by an increase to 220 °C at a rate of 5 °C/min, and finally to 250 °C at a rate of 15 °C/min. The split ratio was 10:1, and the solvent delay was 3.5 min. Data were processed using LabSolution software (version 4.50). Metabolite identification was based on comparing the retention time and specific ion fragments to those within NIST (version 2017). For the discriminant metabolites, the identification was further validated by analyzing standard substances.

LC-MS-Based Analysis in Lung Tissue: For LC-MS analysis, 50 mg of lung tissue was homogenized in 500 μL of MeOH/H_2_O (4:1, *v*/*v*) containing the following internal standards: Carnitine (C2:0), 0.16 μg/mL; LPC (19:0), 0.75 µg/mL; FFA (18:0-d3), 2.5 µg/mL; 2-Chloro-D-phenylalanine, 0.3 µg/mL; and CA-d5, 1.8 µg/mL. The homogenate was vortexed for 2 min. After homogenization, the mixture was thoroughly shaken for 5 min, incubated on ice for 15 min, and then centrifuged at 15,000 rpm for 15 min. The supernatant was dried under nitrogen, reconstituted in 100 µL of acetonitrile/water (1:1, *v*/*v*), and filtered through a 0.22 µm membrane prior to LC-MS analysis. The metabolites were separated on an ACQUITY UPLC HSS T3 column (1.8 µm, 2.1 × 100 mm, Waters, Milford, MA, USA). The column temperature was maintained at 40 °C. In positive mode, the mobile phase was as follows: Mobile phase A consisted of pure water plus 0.1% formic acid, and mobile phase B consisted of acetonitrile plus 0.1% formic acid. Phases A and B for the negative mode were water and 95% MeOH/H_2_O, both with 6.5 mM ammonium bicarbonate, respectively. The flow rate was set to 0.20 mL/min. The injection volume was 10 µL for each analysis. The capillary temperature was maintained at 250 °C, with a capillary voltage of 42 V and a source voltage of 5 kV. The sheath gas was nitrogen.

Lipidomics Analysis: For lipid analysis, 50 mg of lung tissue was mixed with 400 µL of precooled extraction solvent (methanol/dichloromethane, 1:2, *v*/*v*) containing a cocktail of lipid internal standards: PE (17:0/17:0), 0.99 µg/mL; LPC (19:0), 0.33 µg/mL; SM (d18:1/12:0), 0.17 µg/mL; TG (17:0/17:0), 0.53 µg/mL; Cer (d18:1/17:0), 0.51 µg/mL; FFA (18-d3), 0.67 µg/mL; and PC (19:0), 0.67 µg/mL. The homogenate was vortexed for 2 min. After homogenization, the mixture was thoroughly shaken for 5 min, incubated on ice for 15 min, and then centrifuged at 12,000 rpm for 15 min. After centrifugation, the lower organic phase was extracted and dried under nitrogen at 37 °C. The residue was reconstituted in a mixture of isopropanol and methanol (1:1, *v*/*v*) and filtered prior to analysis. An Acclaim C30 column (3.0 µm, 2.1 × 100 mm; Thermo Scientific, Waltham, MA, USA) was used with a column temperature at 45 °C. The flow rate was set to 0.26 mL/min. Mobile phase A was composed of acetonitrile/water (60/40, *v*/*v*, containing 0.1% formic acid, 0.1% ammonium formate); mobile phase B was a mixture of isopropanol/acetonitrile/water (90/10/5, *v*/*v*, containing 0.1% formic acid, 0.1% ammonium formate). The mass spectrometry setup utilized an electrospray ionization (ESI) source with 3800 V for positive mode and 3200 V for negative mode. The capillary temperature was maintained at 320 °C. Mass scans ranged from 100 to 1500 m/z, with a first-level resolution of 70,000 full width at half maximum (FWHM) and a second-level resolution of 17,500 FWHM. Internal standards were used for retention time alignment, peak area normalization, and absolute quantification of lipid species.

Data Processing and Qualitative Analysis: LC-MS data were processed both qualitatively and quantitatively using Thermo Fisher’s Compound Discoverer 3.0 software. The obtained data of MS^1^ and MS^2^ (obtained primarily with several quality control (QC) samples), as well as the retention time of the ions, were identified according to an integrated metabolite/lipid database encompassing the software’s own database and an in-house database established with standard substances and their MS^1^ and MS^2^ information [37]. The identified metabolites were further confirmed using Kyoto Encyclopedia of Genes and Genomes (KEGG, https://www.kegg.jp/ (accessed on 5 December 2024)) and MassBank (mass bank https://massbank.jp/ (accessed on 22 November 2024)). Peak areas from the raw data of the QC samples (subpacked from the QC pool, which was formed by extracting an equal volume of each real sample) were subjected to intra-group variability filtering. Ions detected with more than 10% missing values and relative standard deviations (RSDs) greater than 30% within all the QC samples were subsequently discarded. Metabolomic data were further analyzed using the open-access online platform MetaboAnalyst 6.0 (https://www.metaboanalyst.ca/ (accessed on 7 December 2024)). Principal component analysis (PCA) was used to compare the metabolome and lipidome between groups. The significance of differences between groups was quantified using the PERMANOVA test from the vegan package in R, with Euclidean distance as the distance metric and 999 permutations. Welch’s t-test was performed for differential analysis between two groups. Metabolites with a variable importance in projection (VIP) score ≥ 1 and a *p*-value < 0.05 were considered differential metabolites. Pathway enrichment analysis was conducted on the significantly differential metabolites to explore their associated metabolic pathways.

### 2.10. Reverse Transcription Quantitative Polymerase Chain Reaction (RT-qPCR)

Total RNA was extracted from lung tissues using a total RNA extraction kit, and the RNA concentration was measured. The RNA was then reverse transcribed into complementary DNA (cDNA) using a reverse transcription kit. The mRNA levels of the target genes were measured by RT-qPCR. The primer sequences used for the RT-qPCR were as follows:

Rat-G*pat3* forward: GAACCCGGATGGATCTGCCCA;

Reverse: GTAACACCCAGCCAGTCAGCC.

Rat-S*irt1* forward: GACAGTTCCAGCCATCTCTGT;

Reverse: GCAAGATGCTGTTGCAAAGG. The primers were synthesized by Sangon Biotech (Shanghai, China). The expressions of the genes were represented by the 2^−ΔΔCt^ method.

### 2.11. Western Blotting

Western blotting assays were used to characterize protein expressions in rat lung tissue and BEAS-2B cells. After homogenization, a BCA (bicinchoninic acid) protein assay kit was used to determine protein concentrations in the supernatants. Forty micrograms of each sample was loaded onto an SDS-PAGE (sodium dodecyl sulfate polyacrylamide gel electrophoresis) gel and was transferred to a PVDF membrane, blocked with 5% skim milk. Next, the membrane was incubated with the primary antibodies overnight at 4 °C. The membranes were incubated with a secondary antibody, HRP-conjugated goat anti-rabbit (horseradish peroxidase, HRP), for 1 h at 37 °C. Visualization was performed using a gel imager, and quantification was carried out using ImageJ software (version 1.54d).

### 2.12. Statistical Analysis

Data were expressed as intergroup average ± standard error of the mean (SEM). Multiple group comparisons were analyzed by one-way analysis of variance (ANOVA) followed by Tukey’s post hoc test. Statistical significance was confirmed as *p* < 0.05 was considered as the threshold of significant difference.

## 3. Results

### 3.1. Chiglitazar Improved the Survival Rate of CLP Rats and Reduced Inflammatory Cytokines in BALF

To evaluate the effect of chiglitazar (Figure 2A) on sepsis-related mortality, we tracked the longevity of rats for 72 h post-CLP surgery and constructed a survival curve based on the survival counts across different groups, as shown in Figure 2B. Figure 2C shows that the fatal events in the CLP-induced rats took place between 6 and 24 h after CLP. The CLP group exhibited significantly reduced survival rates at 72 h. Rats pretreated with various doses of chiglitazar (2.5, 5, and 10 mg/kg) showed improved survival, suggesting that chiglitazar could protect rats from CLP-induced sepsis lethality. Furthermore, inflammatory cytokines in the BALF of the rats were quantified. The results demonstrated that, compared with the CLP group, chiglitazar administration significantly reduced the levels of several inflammatory cytokines, including IL-1β, IL-6, and TNF-α (Figure 2D–F). These cytokines play a crucial role in sepsis, with IL-1β and TNF-α being pro-inflammatory factors that exacerbate systemic inflammation, while IL-6 is closely involved in the acute-phase response and immune modulation. By suppressing the production of these cytokines, chiglitazar may alleviate the excessive inflammatory responses in CLP rats.

### 3.2. Chiglitazar Ameliorated Acute Lung Injury in Septic Rats

H&E staining was employed to assess the protective effect of chiglitazar on lung injury in septic rats. Compared to the Sham group, the CLP group exhibited thickening of the alveolar interstitium, extensive aggregation of neutrophils, and widespread infiltration of inflammatory cells within the alveolar lumen. In contrast, the CLP+Pio group (positive control) showed alveolar wall thickening and mild inflammatory cell infiltration, but these changes were significantly milder than those in the CLP group. Similarly, the CLP+Chi group displayed moderate alveolar wall thickening with minimal inflammatory cell infiltration, and the protective effect was dose-dependent (Figure 3A). Quantification of lung injury scores confirmed that both the CLP+Pio group and the CLP+Chi group had significantly lower damage compared to CLP (Figure 3B). The wet-to-dry weight ratios (W/D ratio) in the CLP group were significantly higher than those in the Sham group, indicating more severe pulmonary edema. Both the CLP+Pio group and the CLP+Chi group showed a significant reduction in lung tissue damage (Figure 3C), and the decrease in W/D ratio in these groups indicated that both pioglitazone and chiglitazar effectively alleviated pulmonary edema and improved lung tissue integrity. Immunohistochemical analysis (Figure 3D–F) revealed a reduction in the expression of tight junction proteins Occludin and ZO-1 in the lung barrier of the CLP group. These proteins are critical for maintaining epithelial barrier integrity, and their downregulation indicates impaired lung barrier function. In contrast, the expression of these permeability markers was restored in both the CLP+Pio group and the CLP+Chi group. Overall, the findings suggested that chiglitazar, like pioglitazone, may help reduce pulmonary inflammation and mitigate acute lung injury induced by sepsis.

### 3.3. Chiglitazar Induced the Reprogramming of Kynurenine–Nicotinamide Pathway in Septic Rats

Metabolomic analyses of rat lung tissue samples were performed using LC-MS and GC-MS. The PCA score plots revealed distinct clustering of the Sham, CLP, and CLP+Chi 10 (10 mg/kg chiglitazar) groups, indicating considerable metabolic alterations between the groups and a shifted metabolome mediated by chiglitazar towards that of the controls (Figure 4A). With the thresholds VIP > 1, *p*-value < 0.05, and fold change (FC) ≥ 2, volcano plots identified 41 metabolites with significant differences between the Sham and CLP groups, and 23 metabolites with significant differences between the CLP+Chi 10 and CLP groups (Appendix A and Table 1 and Table 2). Figure 4B,C display the pathway enrichment results for these metabolites, where nicotinate and nicotinamide metabolism were determined as the primary affected pathway by both the CLP operation and the pretreatment of chiglitazar. Notably, the Venn diagram summarizing the overlapped discriminant metabolites between the comparisons Sham vs. CLP and CLP vs. CLP+Chi 10 revealed 11 differential metabolites (Figure 4D). Among these, 10 metabolites exhibited dysregulations in the sepsis model group and were partially restored in the CLP+Chi 10 group (Table 3). Figure 4E–I show the intergroup differences in these metabolites, suggesting that chiglitazar may modulate sepsis-related metabolic disturbances, potentially contributing to its therapeutic effects in sepsis. Specifically, comprehensive dysregulations in the metabolites involved in kynurenine metabolism, summarized in Figure 4J, were observed to be dysregulated in the septic rats but restored in the CLP+Chi 10 group rat lung tissue. Notably, enhanced NAD+, which is known to be a crucial coenzyme for energetic metabolism, was observed following modulation by chiglitazar in septic rats. Such an effect could be attributed to increased levels of 3-hydroxy-L-kynurenine (3-HK) and quinolinate, which are upstream precursors of NAD+ synthesis. Such results are consistent with previous findings that NAD+ could be activated by PPAR agonists [38].

### 3.4. Considerable Dysregulations of Lipid Metabolism Found in Septic Rats Were Reversed by the Pretreatment of Chiglitazar

Lipidomic analysis of lung tissue samples revealed significant alterations in lipid metabolites between the Sham and CLP groups, and the CLP and CLP+Chi 10 groups. The PCA score plots, along with the volcano plots, are shown in Appendix A. A Venn diagram in Appendix A summarizes the overlapped discriminant lipids responsible for both the comparison between the Sham and CLP groups and that between the CLP and CLP+Chi 10 groups. Notable differences were observed in the triglycerides (TGs) and diglycerides (DGs) (Figure 5A–J), which were downregulated in the CLP group compared to the Sham group, while phosphatidylcholines (PCs), lysophosphatidylcholines (LPCs) (Figure 5K,L), and fatty acids (FAs) were upregulated in the CLP group (Figure 5M–P). Furthermore, variations in other differential lipid metabolites, such as sphingomyelins (SMs) and cholinesterases (ChEs), are detailed in Appendix A. Correspondingly, the pretreatment with chiglitazar reversed the variations caused by CLP, restoring the levels of TGs and DGs to those similar to the Sham group, while also decreasing the levels of PCs, LPCs, and FAs that were elevated in the CLP group. In particular, septic rats exhibited remarkable decrements in TGs and DGs (32:0) and increments in FAs compared to the Sham animals. In contrast, the CLP+Chi 10 group showed significant increases in TGs and DGs (32:0), along with decreases in FAs, compared to the CLP group. These shifts suggest that chiglitazar might enhance the conversion of FAs into TGs during sepsis, thereby restoring lipid metabolism by limiting excessive free fatty acid accumulation and promoting the esterification of FAs into more stable lipid forms. Such metabolic changes are likely to alleviate oxidative stress in mitochondria, thereby improving overall cellular energy supply and tissue injury.

### 3.5. Chiglitazar Pretreatment Ameliorates Sepsis-Induced Metabolic Dysregulation via Activation of the SIRT1/PGC-1α/PPARα/GPAT3 Signaling Axis

SIRT1 catalyzes the deacetylation-dependent hydrolysis of NAD+ into nicotinamide (NAM) [39]. Elevated NAD+ levels in chiglitazar-pretreated CLP rats could arise from reduced SIRT1 activity (slowing NAD+ breakdown) or increased SIRT1 expression. However, the unchanged NAD+/NAM ratio between the CLP and CLP+Chi 10 groups (Figure 6A,B) excludes altered SIRT1 activity, as enhanced activity would lower this ratio via faster NAD+ conversion to NAM. Instead, restored NAD+ levels likely result from upregulated SIRT1 expression, which improves NAD+ salvage pathways (e.g., nicotinamide phosphoribosyltransferase (NAMPT)-mediated regeneration) without directly modifying enzymatic activity [40]. This aligns with studies showing that elevated SIRT1 protein stabilizes NAD+ availability through salvage pathways, even without hyperactivity [41]. Meanwhile, the lipidomic profiling showed that, compared to the CLP group, the intervention with chiglitazar led to remarkable conversions of excess FAs into several TGs and one DG, highlighting the crucial role of glycerol-3-phosphate acyltransferase 3 (GPAT3), known as a rate-limiting enzyme in the conversion of FAs into TGs and DGs. To investigate whether they were targeted by chiglitazar, we evaluated the mRNA and protein expressions of SIRT1 and GPAT3 in the lung tissues from the modeled rats. Given the suboptimal efficacy of the CLP+Chi 2.5 mg/kg dose observed in prior in vivo experiments, we selected the CLP+Chi 5 mg/kg and CLP+Chi 10 mg/kg doses for mechanistic validation. As shown in Figure 6C,D, both mRNA levels of *Sirt1* and *Gpat3* were significantly reduced in the CLP group compared to the Sham group. However, these levels were restored in the CLP+Chi 10 group, suggesting an improvement in sepsis-associated lung injury. Interestingly, while SIRT1 is recognized for its indirect regulatory roles in lipid metabolism, existing evidence does not support direct transcriptional or post-translational control of GPAT3 by SIRT1. We therefore posited that intermediary targets mediate SIRT1’s influence on GPAT3. Furthermore, PPARα directly activates GPAT3 transcription [42], and NAD+-dependent SIRT1 activation enhances PGC-1α/PPARα heterodimerization [43]. Collectively, these findings support the hypothesis that chiglitazar alleviates SALI by orchestrating a hierarchical SIRT1/PGC-1α/PPARα/GPAT3 signaling cascade, wherein SIRT1 indirectly regulates GPAT3 via PGC-1α-mediated PPARα activation. The Western blot analysis (Figure 6E,F) further confirmed that the protein levels of SIRT1 and GPAT3 were significantly increased in the CLP+Chi 10 group compared to the CLP group (Figure 6G,H), consistent with their mRNA expression. Similarly, chiglitazar treatment resulted in notable upregulation of PGC-1α and PPARα protein expressions (Figure 6I,J), indicating the involvement of these regulatory pathways in the observed therapeutic effects. Additionally, unlike the significant inter-group differences in the gene expressions of *Sirt1* and *Gpat3* between the CLP rats treated with two doses of chiglitazar, no significant variations were found in the protein expressions of SIRT1, GPAT3, PGC−1α, and PPARα between the two groups.

### 3.6. SIRT1 Inhibitor EX-527 Reversed the Activation of Signaling Pathways by Chiglitazar in BEAS-2B Cells

To further investigate the role of SIRT1 in mediating the effects of chiglitazar in vitro, its cytotoxicity in BEAS-2B cells was first assessed using the CCK-8 assay. Cells were treated with various doses of chiglitazar for 24 h, and no significant cytotoxicity was observed at concentrations below 64 μM (Appendix A). At 32 μM, chiglitazar significantly increased cell viability compared to the LPS-treated group, whereas higher concentrations decreased cell viability. Based on these results, 8, 16, and 32 μM were selected as the low, medium, and high doses for subsequent experiments. Western blot analysis showed a significant increase in the protein levels of SIRT1, PGC-1α, PPARα, and GPAT3 in BEAS-2B cells from the CLP+Chi group compared to the CLP group. To determine whether high expression of SIRT1 was essential for the observed effects, the SIRT1 inhibitor EX-527 was utilized to evaluate the expression of the downstream targets involved in lipid metabolism and mitochondrial protection. Consequently, co-administration of EX-527 significantly suppressed the chiglitazar-induced activation of the SIRT1/PGC-1α/PPARα/GPAT3 axis (Figure 7A–E). These results indicate that the inhibition of SIRT1 effectively reversed the regulatory effects of chiglitazar on this signaling pathway, thereby mitigating its action on sepsis-induced lung injury. Furthermore, inflammatory cytokine levels (IL-6, IL-1β, TNF-α) in LPS-induced BEAS-2B cells were significantly reduced following chiglitazar treatment (Appendix A).

### 3.7. The SIRT1 Inhibitor EX-527 Reversed the Effect of Chiglitazar on Sepsis-Induced Lung Injury

To confirm the reversal of chiglitazar’s effects by EX-527 in vivo, EX-527 and chiglitazar were co-administered to a separate group of CLP rats. Compared to the rats pretreated with chiglitazar alone, co-administration of EX-527 significantly decreased survival rates in the CLP model (Figure 8A). Histological examination with H&E staining revealed characteristic alveolar thickening and inflammatory cell infiltration in the CLP group. In contrast, the CLP+Chi 10 group showed alleviation of these pathological changes, while co-administration of EX-527 and chiglitazar exacerbated lung injury (Figure 8B). The W/D ratio and lung injury scores further confirmed the detrimental effects of EX-527 on lung injury (Figure 8C,D). Immunohistochemical analysis (Figure 8E–G) showed that the expression of tight junction proteins, Occludin and ZO-1, was restored in the CLP+Chi 10 group but reversed by EX-527, indicating the loss of chiglitazar’s protective effect. Western blot analysis (Figure 8H–L) revealed significant increases in the protein levels of SIRT1, PGC-1α, PPARα, and GPAT3 in the CLP+Chi 10 group compared to the CLP group. As expected, co-administration of EX-527 effectively nullified the regulatory effects of chiglitazar on this signaling pathway in vivo. Furthermore, ELISA results (Appendix A) showed that the anti-inflammatory effect of chiglitazar was blocked by the SIRT1 inhibitor EX-527, resulting in increased cytokine levels. Collectively, these findings support the conclusion that chiglitazar mitigates sepsis-induced lung injury through the SIRT1/PGC-1α/PPARα/GPAT3 signaling pathway. Figure 8M presents a schematic summary of our findings.

## 4. Discussion

Dysregulation of metabolic processes, particularly aberrations in energy metabolism, represent a key hallmark of sepsis progression. Increasing evidence indicates that correcting dysregulated energy metabolism ameliorates sepsis-induced organ dysfunction, highlighting its potential as a therapeutic target for sepsis management [20]. Our study demonstrates metabolic resuscitation as a viable therapeutic paradigm by demonstrating that chiglitazar, a PPAR pan-activator, orchestrates mitochondrial-lipid homeostasis to protect against SALI. While prior PPAR research predominantly focused on immunomodulation [29,44], our multi-omics dissection reveals that restructuring metabolic networks forms the mechanistic basis of chiglitazar’s organ protection through the SIRT1/PGC-1α/PPARα/GPAT3 axis.

Histopathological validation showed that chiglitazar preserves pulmonary endothelial integrity, as evidenced by restored Occludin/ZO-1 expression. This aligns with PPARα’s known vascular protective roles [45,46], yet extends the mechanistic understanding to mitochondrial-lipid crosstalk. Notably, pioglitazone, despite being a PPARγ-dominant agonist with minor PPARα activation, exhibited effects comparable to chiglitazar in mitigating lung injury. This observation reflects pioglitazone’s partial PPARα agonism at therapeutic doses [47,48], which may synergize with its anti-inflammatory PPARγ activity to modulate lipid metabolism in SALI. Furthermore, as the only clinically approved thiazolidinedione for sepsis-related trials [49], pioglitazone serves as a pharmacologically relevant benchmark, even though its mechanism diverges from chiglitazar’s pan-PPAR activation. These findings suggest that PPARγ-mediated inflammation resolution and PPARα-driven lipid remodeling may cooperatively protect against SALI, highlighting the therapeutic potential of multitargeted PPAR modulation.

Metabolomic profiling identified NAD+ depletion as a pivotal driver of bioenergetic collapse in sepsis. NAD+ is a crucial coenzyme involved in cellular energy production, particularly within the tricarboxylic acid (TCA) cycle and the electron transport chain [50]. In sepsis, NAD+ depletion has been associated with impaired energy metabolism and exacerbated organ dysfunction [51]. Our study highlights that chiglitazar restores NAD+ levels, suggesting its action was not through deacetylation-associated activation but through the increase in the expression of SIRT1, a critical modulator of anti-inflammatory and energy supply-related signaling [41]. Importantly, SIRT1 upregulation is thought to enhance NAD+ consumption (via the conversion to NAM). However, this hypothesis is not supported by our findings as neither significant NAM accumulation nor altered NAD+/NAM ratios were observed. Crucially, while chiglitazar elevates SIRT1 expression, the restored NAD+ levels are likely prioritized for mitochondrial bioenergetics rather than SIRT1-dependent deacetylation [52,53]. This suggests that chiglitazar alleviates sepsis-induced NAD+ depletion not by merely compensating for SIRT1 substrate demand in the cytoplasm but through metabolic reprogramming that redirects NAD+ toward sustaining mitochondrial respiration and redox homeostasis. These observations align with the critical role of NAD+ in mitochondrial energy metabolism, further supporting chiglitazar’s therapeutic potential in resolving sepsis-associated bioenergetic failure. Additionally, the dampened synthesis of NAD+ is also associated with kynurenine accumulation during sepsis, as evidenced in the current study. Interestingly, increased kynurenine is implicated in inflammatory responses and immune dysfunctions [54,55]. Therefore, our findings imply that the anti-inflammatory effect of chiglitazar may be partly attributable to its kynurenine suppression actions. Literally, our findings suggest that chiglitazar does decrease the levels of inflammatory cytokines, including IL-1β, IL-6, and TNF-α, in the BALF of septic rats, confirming its role in suppressing systemic inflammatory responses. This is consistent with the work of Cheng et al., who reported that the activation of PPARγ significantly reduced the levels of inflammatory factors in septic mice by inhibiting the NF-κB signaling pathway [56].

Lipidomic data revealed that another remarkable effect of chiglitazar in modulating metabolic dysregulation is the promotion of the depletion of excessive fatty acids through the upregulation of GPAT3 expression. It has been documented that overexpression of GPAT3 in mammals significantly increases TG formation [57]. Of note, raised TGs in sepsis are thought to be associated with a reduction in severity and improved outcomes of SALI due to their antioxidant effects and TG-associated regulator T cell activation [58,59]. As a PPARα agonist, chiglitazar is known to activate the β-oxidation of fatty acids in diabetic patients. Beyond that, our findings indicate that chiglitazar promotes the conversion of excess FAs into TGs, which may further help prevent the accumulation of fatty acids around pulmonary mitochondria. Such effects reduce lipotoxicity and support mitochondrial integrity and thereby improve dysregulated energy metabolism and energy supply-associated lung injuries. Notably, the reduction in FFAs may also alleviate mitochondrial oxidative stress by decreasing lipid peroxidation (thereby limiting ROS generation) and restoring electron transport chain (ETC) efficiency. Excessive FFAs are known to induce ROS overproduction and impair ETC function [60,61,62]. However, given the multifaceted roles of PGC-1α in various mitochondrial injury-associated pathways, these mechanisms, while potentially contributory, are likely not the primary drivers of the observed effects. Instead, the primary mechanism underlying the improvements in mitochondrial function and overall metabolic health in our study is the metabolic reprogramming mediated by the SIRT1/PGC-1α/PPARα/GPAT3 axis. Interestingly, as a marker of mitochondrial biogenesis, PGC-1α is implicated with the improved oxidative phosphorylation capacity of mitochondria. The increased expression in PGC-1α induced by chiglitazar also evidences its action on mitochondrial protection and cellular homeostasis [63].

Despite these advances, several limitations remain to be considered in the current work. First, sepsis is a highly complex and dynamic syndrome leading to fluctuating immune responses, which may provoke uncertain metabolic alterations during the progression of sepsis [64]. As a result, the determinations of optimal time points for both the metabolic intervention and the time of sampling become challenging. In this study, we adopted a preventive treatment strategy, administering chiglitazar before the onset of sepsis in order to avoid gavage-associated secondary harm to the experimental animals after CLP modeling. However, this approach may not fully capture the effects of the drug during the later stages of sepsis, with the presence of immunosuppression and possible opposite metabolic features to those of our findings. Future studies should explore the effects of chiglitazar at different stages of sepsis to better understand the timing of drug administration after the onset of sepsis. Secondly, chiglitazar is a multi-agonist of PPARs. All three receptor subtypes (PPARα, PPARβ/δ, and PPARγ) are known to exhibit anti-inflammatory effects. Hence, it is difficult to discern the specific contribution of each subtype to its overall anti-inflammatory action. Finally, although chiglitazar has been widely utilized clinically for glycemic control, no previous work represents its first application in sepsis. Therefore, additional clinical trials are indispensable to ascertain its safety and efficacy before its application in the management of human sepsis.

In conclusion, this research evidences the value of chiglitazar for the improvement of SALI. By focusing on repairing the mitochondrial lipid axis instead of suppressing the immune system, chiglitazar serves as a prime example of metabolic therapy in sepsis. The effective use of multi-omics phenotyping highlights the crucial role of systems biology in unraveling intricate therapeutic mechanisms.

## Figures and Tables

**Figure 1 metabolites-15-00290-f001:**
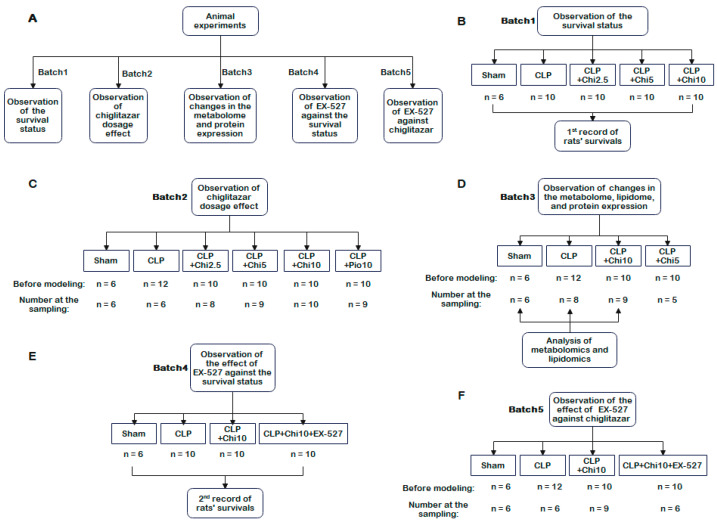
Groups of animal experiments. (**A**) Groupings of five batches of experimental animals. (**B**) Batch 1: observation of the survival status. (**C**) Batch 2: observation of chiglitazar dosage effect. (**D**) Batch 3: observation of changes in the metabolome, lipidome, and protein expression. (**E**) Batch 4: observation of the effect of EX-527 against the survival status. (**F**) Batch 5: observation of the effect of EX-527 against chiglitazar. Sham: The sham-operation group underwent laparotomy and suture only and served as the control. CLP: The sepsis model group underwent cecal ligation and puncture surgery. CLP+Chi 2.5/5/10: Sepsis model rats pretreated with chiglitazar at 2.5, 5, or 10 mg/kg (denoted as CLP+Chi 2.5, CLP+Chi 5, and CLP+Chi 10, respectively). CLP+Pio 10: The group of CLP rats pretreated with pioglitazone (CLP+Pio 10 mg/kg). CLP+Chi+EX-527: both EX-527 (5 mg/kg) and chiglitazar (10 mg/kg) were given to the CLP+Chi 10+EX-527 group.

**Figure 2 metabolites-15-00290-f002:**
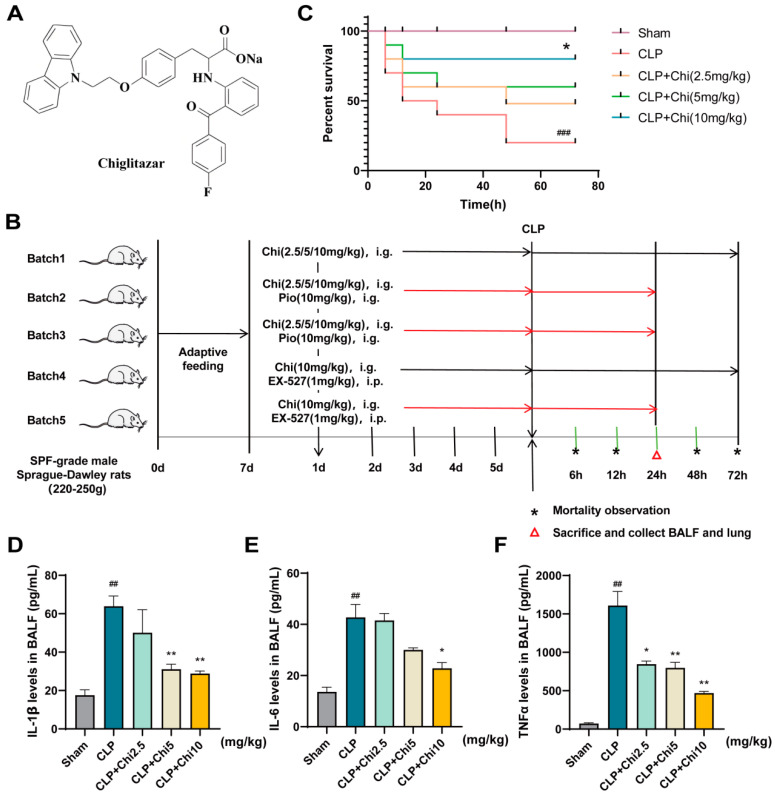
Chiglitazar improved survival in septic rats and reduced the levels of inflammatory cytokines. (**A**) Structural diagram of chiglitazar. (**B**) Workflow for the operations on CLP-induced septic rats with chiglitazar. (**C**) SPF-grade male SD rats were pretreated daily with chiglitazar (2.5, 5, and 10 mg/kg, i.g.), and their survival was monitored for 72 h after modeling operations. (**D**–**F**) Representations of the measurements of IL–1β (**D**), IL–6 (**E**), and TNF–α (**F**) in the BALF of rats. Data represent mean ± SEM (*n* = 3). ^##^ *p* < 0.01 vs. Sham group; ^###^ *p* < 0.001 vs. Sham group; * *p* < 0.05, ** *p* < 0.01 vs. CLP group.

**Figure 3 metabolites-15-00290-f003:**
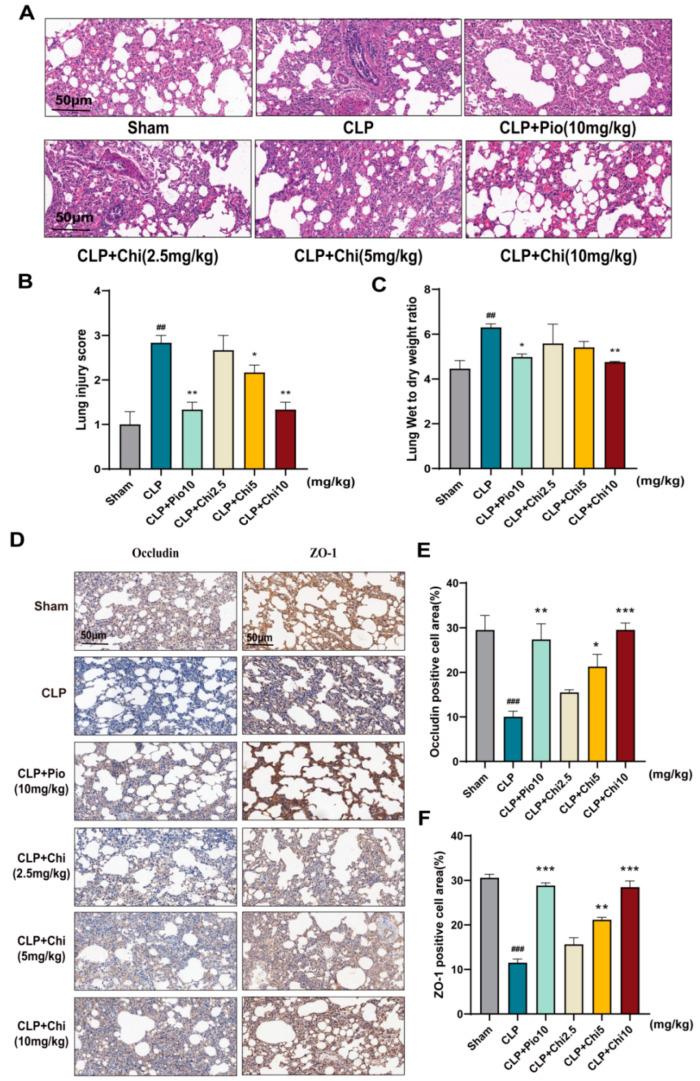
Chiglitazar ameliorated acute lung injury in septic rats. (**A**) Representative histological images of lung tissue from each group stained with H&E. (**B**) Morphological inflammation scores for lung tissue from each group. (**C**) The W/D ratio of lung tissue in rats. (**D**) Immunohistochemical images showing the expression of tight junction proteins ZO-1 and Occludin in the rat lung epithelial barrier. (**E**,**F**) The percentage of positive cell area for Occludin and ZO-1. Data represent mean ± SEM (*n* = 3). ^##^ *p* < 0.01, ^###^ *p* < 0.001 vs. Sham group; * *p* < 0.05, ** *p* < 0.01, *** *p* < 0.001 vs. CLP group.

**Figure 4 metabolites-15-00290-f004:**
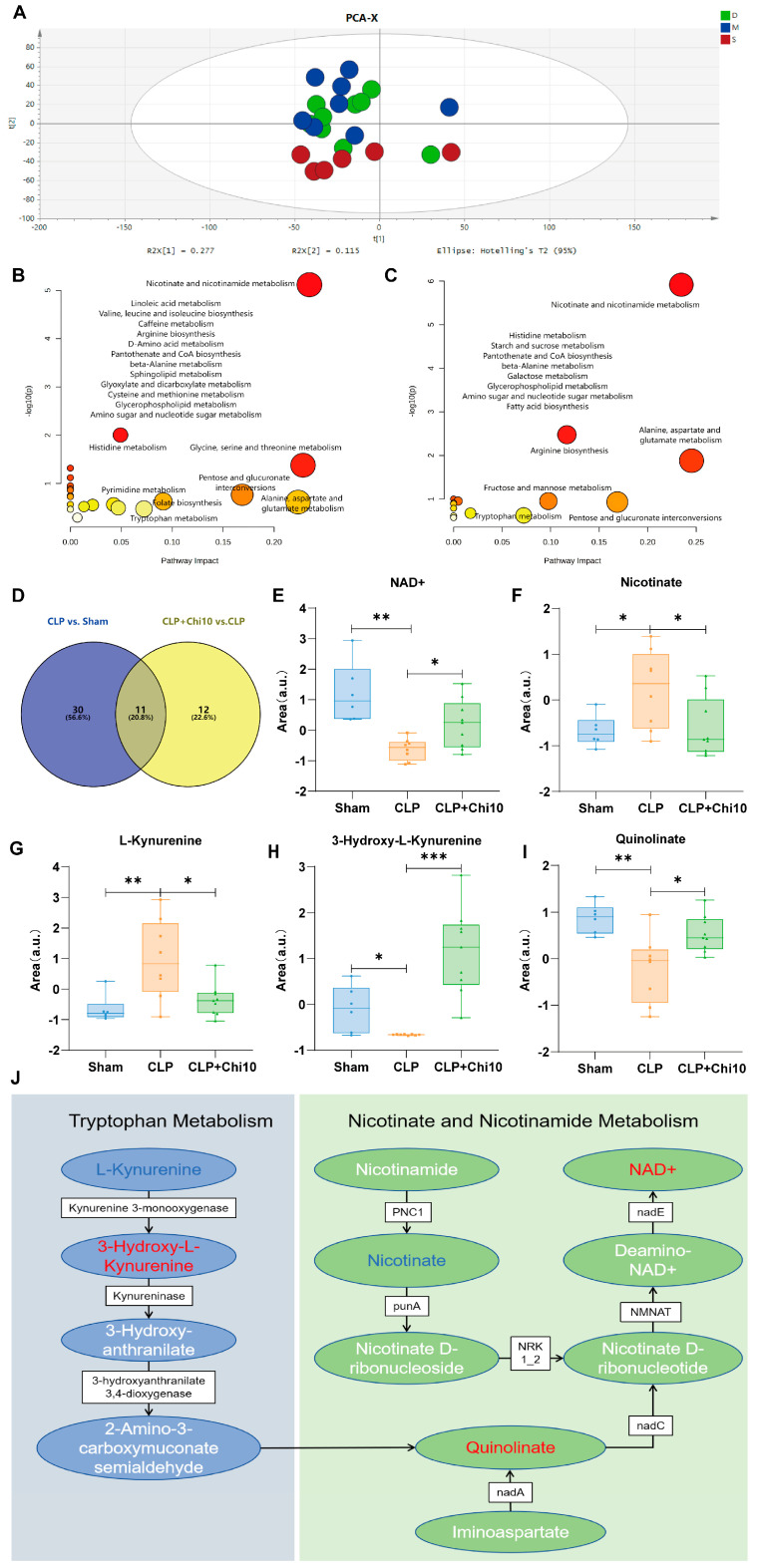
Chiglitazar induced the reprogramming of kynurenine–nicotinamide pathway in septic rats. (**A**) PCA of rat lung tissue metabolomics analysis between groups, where “S” represents the Sham surgery group; “M” represents the CLP group; “D” represents the CLP+Chi (10 mg/kg) group. Ninety-five percent confidence ellipses (multivariate normal distribution) illustrate group separation trends in principal component space. Inter-group dissimilarities were assessed by PERMANOVA (Euclidean distance matrix, 999 permutations; F = 2.881, *p* = 0.002). (**B**,**C**) Pathway enrichment results for CLP vs. Sham and CLP+Chi 10 vs. CLP discriminant analyses. (**D**) Venn diagram showing the differential metabolites between the CLP vs. Sham and CLP+Chi 10 vs. CLP groups. (**E**–**I**) Intergroup differences in metabolite levels detected in the kynurenine and nicotinic acid partial metabolic pathways. (**J**) Schematic diagram of the metabolic pathway involving nicotinic acid and kynurenine-related metabolites impacted by chiglitazar. Metabolites highlighted in red are those that increased in the CLP+Chi 10 group compared to the CLP group; the blue color indicates decreased levels, and white represents metabolites that were undetected in the analysis. * *p* < 0.05; ** *p* < 0.01; *** *p* < 0.001.

**Figure 5 metabolites-15-00290-f005:**
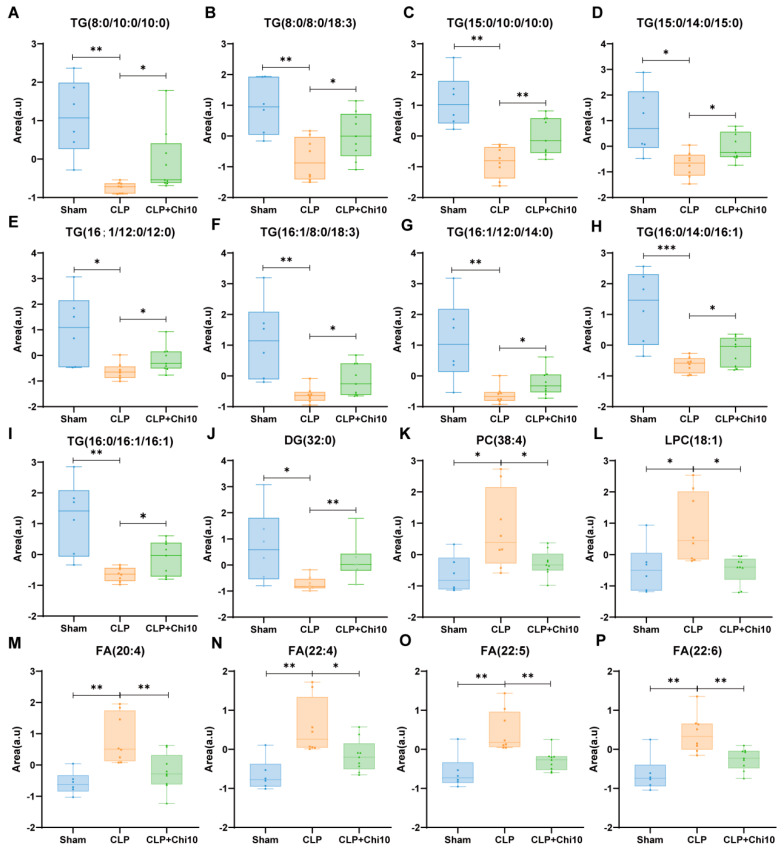
Differential lipid metabolites in the lung tissue of septic rats between groups. (**A**−**I**) Discriminant TGs; (**J**) DG (32:0); (**K**) PCs; (**L**) LPCs; and (**M**−**P**) FAs. * *p* < 0.05; ** *p* < 0.01; *** *p* < 0.001.

**Figure 6 metabolites-15-00290-f006:**
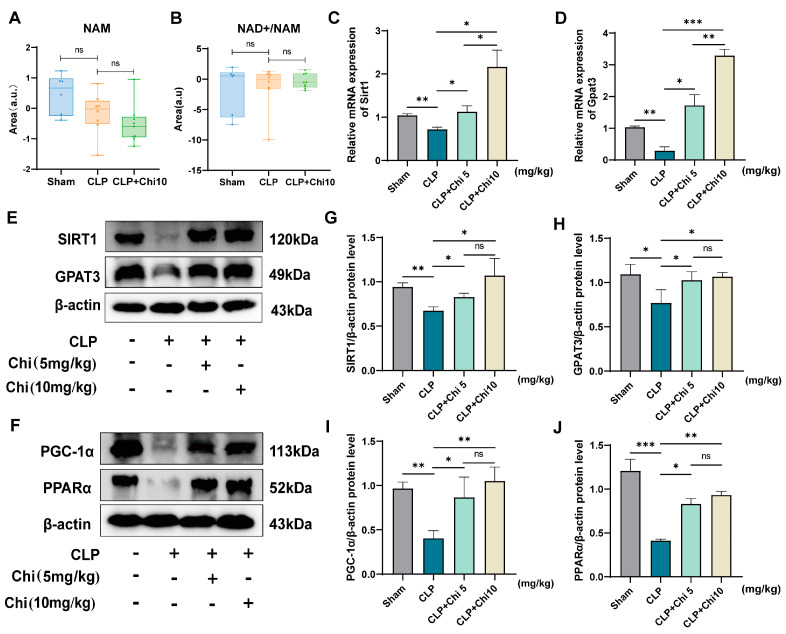
Chiglitazar–mediated lung protection was associated with upregulated SIRT1/PGC−1α/PPARα/GPAT3 signaling pathway in rats. (**A**,**B**) Relative quantification of nicotinamide (NAM) and the ratio NAD+/NAM in rat lung tissue. (**C**,**D**) Relative expression of *Sirt1* and *Gpat3* mRNAs in rat lung tissues. (**E**,**F**) SIRT1, GPAT3, PGC−1α, and PPARα protein expression in rat lung tissues. (**G**−**J**) Quantification of the relative expression levels of SIRT1, GPAT3, PGC−1α, and PPARα proteins. Data represent mean ± SEM (*n* = 3). * *p* < 0.05; ** *p* < 0.01; *** *p* < 0.001; ns indicates no significant differences between groups.

**Figure 7 metabolites-15-00290-f007:**
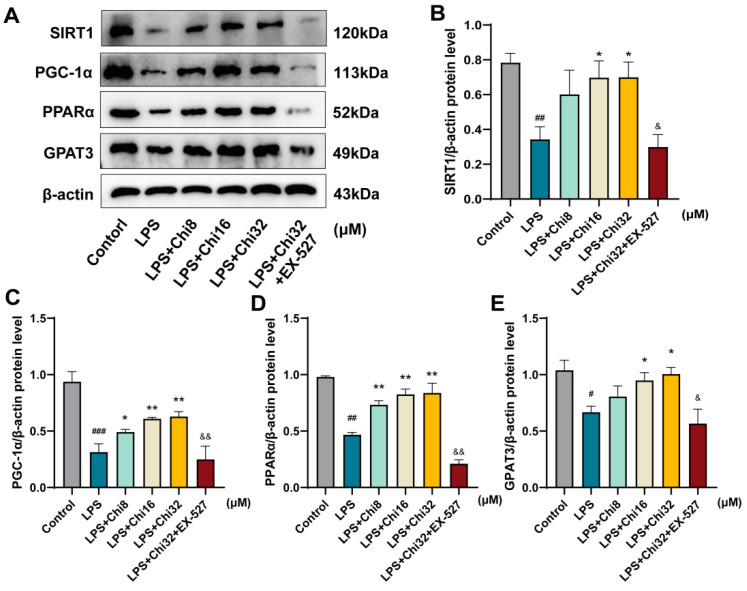
SIRT1 inhibitor EX–527 reversed the upregulation of the PGC–1α/PPARα/GPAT3 signaling by chiglitazar in LPS–stimulated BEAS-2B Cells. (**A**) SIRT1, PGC-1α, PPARα, and GPAT3 protein expressions in BEAS-2B cells. (**B**–**E**) Quantification of the relative expression levels of SIRT1, PGC-1α, PPARα, and GPAT3 proteins. Data represent mean ± SEM (*n* = 3). ^#^ *p* < 0.05, ^##^ *p* < 0.01, ^###^ *p* < 0.001 vs. control group; * *p* < 0.05, ** *p* < 0.01 vs. LPS group; & *p* < 0.05, && *p* < 0.01 vs. LPS+Chi32 group.

**Figure 8 metabolites-15-00290-f008:**
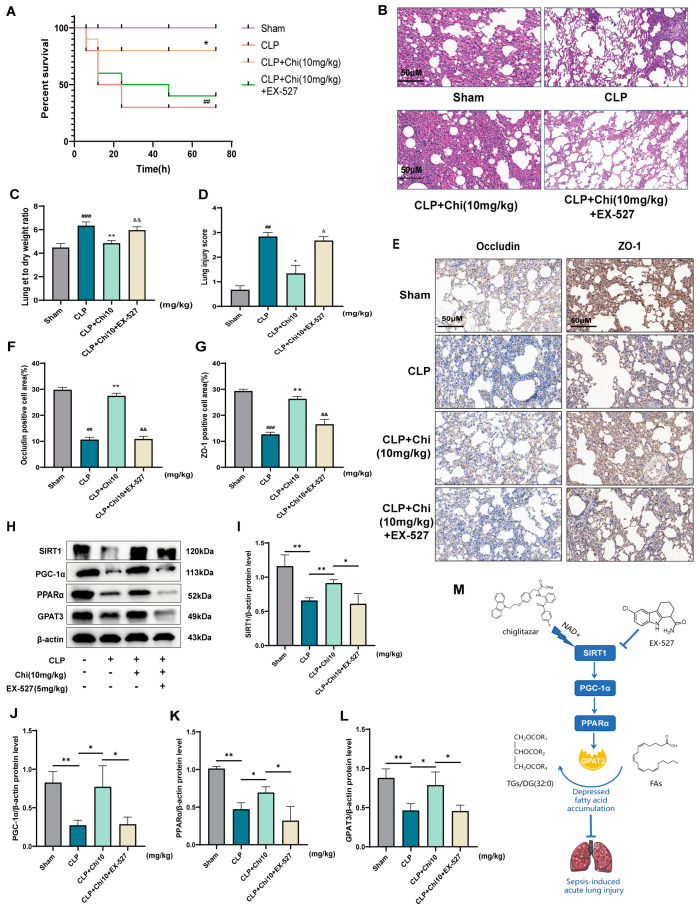
The SIRT1 inhibitor EX-527 reversed the protective effect of chiglitazar on sepsis-induced lung injury. (**A**) Seventy-two-hour survival curve of rats. (**B**) Representative histological images of lung tissue from each group stained with H&E. (**C**) The W/D ratio of lung tissue for each group of rats. (**D**) Morphological inflammation scores for lung tissue. (**E**) Immunohistochemical images showing the expression of ZO-1 and Occludin in lung tissues. (**F**,**G**) Percentages of positive cell area for Occludin (**F**) and ZO-1 (**G**). (**H**) Representative images of SIRT1, PGC-1α, PPARα, and GPAT3 protein expression in lung tissues. (**I**–**L**) Quantification of the relative expression levels of SIRT1, PGC-1α, PPARα, and GPAT3 proteins. (**M**) Schematic illustration of the molecular mechanisms involved in the protective effects of chiglitazar on acute lung injury caused by sepsis-induced metabolic disorders. Data represent mean ± SEM (*n* = 3). * *p* < 0.05; ** *p* < 0.01;. ^##^ *p* < 0.01, ^###^ *p* < 0.001 vs. Sham group; * *p* < 0.05, ** *p* < 0.01 vs. CLP group; ^&^
*p* < 0.05, ^&&^
*p* < 0.01 vs. CLP+Chi10 group.

**Table 1 metabolites-15-00290-t001:** Differential metabolites between CLP and Sham groups. Statistical significance was confirmed as *p* < 0.05 was considered as the threshold of significant difference. *: identification confirmed with in-house database established with standard substance; up/down: increase/decrease in the CLP group.

Metabolite	Formula	m/z	RT (min)	CLP vs. Sham	Mode	Platform
Nicotinate *	C_6_H_5_NO_2_	125.0428	5.191	up	Negative	LC-MS
L-Aspartate *	C_4_H_7_NO_4_	134.0448	1.03	up	Negative	LC-MS
Hydroxyadipic acid *	C_6_H_10_O_5_	143.03385	1.386	down	Negative	LC-MS
N2-gamma-Glutamylglutamine	C_10_H1_7_N_3_O_6_	149.55524	6.81	up	Positive	LC-MS
Orotic acid	C_5_H_4_N_2_O_4_	155.01058	1.423	down	Negative	LC-MS
Quinolinate *	C_7_H_5_NO_4_	168.0291	1.437	down	Negative	LC-MS
N-Acetylisoleucine	C_8_H_15_NO_3_	174.11229	6.78	up	Positive	LC-MS
Glucosamine	C_6_H_13_NO_5_	180.08547	10.982	up	Positive	LC-MS
Methoxytyrosine	C_10_H_13_NO_4_	192.06581	6.82	up	Negative	LC-MS
Iduronic acid	C_6_H_10_O_7_	193.03461	1.425	up	Negative	LC-MS
m-Methylhippuric acid	C_10_H_11_NO_3_	194.08097	6.811	up	Positive	LC-MS
Propionylcarnitine *	C_10_H_19_NO_4_	198.11297	8.392	up	Negative	LC-MS
methylene heptanoylglycine	C_10_H_17_NO_3_	200.12782	8.382	up	Positive	LC-MS
L-Kynurenine *	C_10_H_12_N_2_O_3_	209.0928	6.493	up	Positive	LC-MS
Naphthalenedicarboxylic acid	C_12_H_8_O_4_	215.03224	2.25	down	Negative	LC-MS
3-Hydroxy-L-Kynurenine *	C_10_H_12_N_2_O_4_	225.0872	7.119	down	Positive	LC-MS
Methylacetoacetic acid	C_5_H_8_O_3_	231.08715	5.651	down	Negative	LC-MS
Tetradecanedioic acid	C_14_H_26_O_4_	239.16498	11.746	up	Negative	LC-MS
Leucyl-Aspartate	C_10_H_18_N_2_O_5_	247.12855	5.162	up	Positive	LC-MS
Hexadecanedioic acid *	C_16_H_30_O_4_	267.19659	12.875	up	Negative	LC-MS
L-Thyronine	C_15_H_15_NO_4_	274.1103	5.999	up	Positive	LC-MS
Amino-1-MethylUracil	C_5_H_7_N_3_O_2_	283.11477	6.622	up	Positive	LC-MS
Taurolithocholic acid 3-sulfate	C_26_H_45_NO_8_S_2_	293.64984	9.616	up	Positive	LC-MS
N-octanoylglutamine	C_13_H_24_N_2_O_4_	295.16294	5.71	up	Positive	LC-MS
Epoxyoctadecenoic acid	C_18_H_32_O_3_	295.22779	14.11	up	Negative	LC-MS
N-Acetylneuraminic acid *	C_11_H_19_NO_9_	308.09873	1.463	up	Negative	LC-MS
2-hydroxy-dAMP	C_10_H_14_N_5_O_7_P	346.0631	6.785	up	Negative	LC-MS
Methyluric acid	C_6_H_6_N_4_O_3_	363.07568	6.853	down	Negative	LC-MS
Eicosatrienoylethanolamide	C_24_H_36_O_3_	373.27321	9.817	up	Positive	LC-MS
Linolenylcarnitine *	C_25_H_43_NO_4_	422.32587	11.679	up	Positive	LC-MS
Arachidonoylcarnitine *	C_27_H_46_NO_4_	428.31662	16.531	up	Negative	LC-MS
PI (22:5/18:0)	C_49_H_85_O_13_P	468.2865	9.819	up	Positive	LC-MS
Lignoceroylcarnitine *	C_31_H_62_NO_4_	512.46655	15.03	down	Positive	LC-MS
LPE (22:1/0:0)	C_27_H_54_NO_7_P	536.3343	10.026	down	Positive	LC-MS
LysoPI (20:4/0:0)	C_29_H_49_O_12_P	619.28882	13.46	down	Negative	LC-MS
NAD+	C_21_H_27_N_7_O_14_P_2_	663.3186	6.245	down	Positive	LC-MS
L-Threonine *	C_4_H_9_NO_3_	219	15.34	up	-	GC-MS
Oxalic acid *	C_2_H_2_O_4_	147	7.645	up	-	GC-MS
L-Serine *	C_3_H_7_NO_3_	132	11.275	up	-	GC-MS
Ethanolamine *	C_2_H_7_NO	174	11.54	up	-	GC-MS
Xylitol	C_5_H_12_O_5_	74	30.095	down	-	GC-MS

**Table 2 metabolites-15-00290-t002:** Differential metabolites between CLP+Chi 10 and CLP groups. Statistical significance was confirmed as *p* < 0.05 was considered as the threshold of significant difference. *: identification confirmed with in-house database established with standard substance; up/down: increase/decrease in the CLP+Chi 10 group.

Metabolite	Formula	m/z	RT (min)	CLP+Chi 10 vs. CLP	Mode	Platform
Nicotinate *	C_6_H_5_NO_2_	125.0428	5.191	down	Negative	LC-MS
Methylhistamine	C_6_H_11_N_3_	126.10256	1.341	down	Positive	LC-MS
Myristic acid *	C_14_H_28_O_2_	126.10257	1.224	down	Positive	LC-MS
L-Aspartate *	C_4_H_7_NO_4_	134.0448	1.03	up	Negative	LC-MS
Dihydroxy-2-methylpropionic acid	C_4_H_8_O_4_	143.03091	7.424	down	Positive	LC-MS
Quinolinate *	C_7_H_5_NO_4_	168.0291	1.437	up	Negative	LC-MS
L-Kynurenine *	C_10_H_12_N_2_O_3_	209.0928	6.493	down	Positive	LC-MS
L-Glyceric acid *	C_3_H_6_O_4_	213.06197	12.119	up	Positive	LC-MS
N-Acetyl-L-tyrosine *	C_11_H_13_NO_4_	222.08011	6.287	up	Negative	LC-MS
3-Hydroxy-L-Kynurenine *	C_10_H_12_N_2_O_4_	225.0872	6.119	up	Positive	LC-MS
LysoPC (14:1/0:0)	C_22_H_44_NO_7_P	244.63725	9.148	up	Positive	LC-MS
L-Thyronine	C_15_H_15_NO_4_	274.1103	5.999	down	Positive	LC-MS
2-hydroxy-dAMP	C_10_H_14_N_5_O_7_P	346.0631	6.785	down	Negative	LC-MS
SM (d17:2/20:5-3OH(5,6,15))	C_42_H_73_N_2_O_9_P	389.24638	15.244	down	Negative	LC-MS
Oxocholic acid	C_24_H_38_O_5_	405.26466	9.228	up	Negative	LC-MS
LPE (22:1/0:0)	C_27_H_54_NO_7_P	536.3343	10.026	up	Positive	LC-MS
LysoPI (0:0/18:0)	C_27_H_53_O_12_P	623.31586	5.563	up	Positive	LC-MS
NAD+	C_21_H_27_N_7_O_14_P_2_	663.3186	6.245	up	Positive	LC-MS
Quillaic acid	C_74_H_114_O_39_	812.32325	6.797	down	Negative	LC-MS
PI (18:1/16:0)	C_43_H_81_O_13_P	835.53522	9.843	up	Negative	LC-MS
PI (16:0/18:0)	C_43_H_83_O_13_P	837.54931	9.845	up	Negative	LC-MS
Xylitol	C_5_H_12_O_5_	74	30.095	up	-	GC-MS
Oxalic acid *	C_2_H_2_O_4_	147	7.645	down	-	GC-MS

**Table 3 metabolites-15-00290-t003:** The shared differential metabolites derived from the discriminant analyses of CLP vs. Sham and CLP vs. CLP+Chi 10.

Metabolite	Formula	m/z	RT (min)	CLP vs. Sham	CLP+Chi 10 vs. CLP	Mode	Platform
Quinolinate	C_7_H_5_NO_4_	168.0291	1.437	down	up	Negative	LC-MS
L-Aspartate	C_4_H_7_NO_4_	134.0448	1.030	up	up	Negative	LC-MS
2-hydroxy-dAMP	C_10_H_14_N_5_O_7_P	346.0631	6.785	up	down	Negative	LC-MS
NAD+	C_21_H_27_N_7_O_14_P_2_	663.3186	6.245	down	up	Positive	LC-MS
L-Kynurenine	C_10_H_12_N_2_O_3_	209.0928	6.493	up	down	Positive	LC-MS
Nicotinate	C_6_H_5_NO_2_	125.0428	5.191	up	down	Negative	LC-MS
LPE (22:1/0:0)	C_27_H_54_NO_7_P	536.3343	10.026	down	up	Positive	LC-MS
L-Thyronine	C_15_H_15_NO_4_	274.1103	5.999	up	down	Positive	LC-MS
3-Hydroxy-L-Kynurenine	C_10_H_12_N_2_O_4_	225.0872	7.119	down	up	Positive	LC-MS
Oxalic acid	C_2_H_2_O_4_	147	7.645	down	up	-	GC-MS
Xylitol	C_5_H_12_O_5_	74	30.095	up	down	-	GC-MS

## Data Availability

The involved data for the current study have been uploaded to an open-access document with the DOI: 10.6084/m9.figshare.28143026.

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
