# Peer review of "Integrated Metabolomics and Lipidomics Analysis Reveals the Mechanism Behind the Action of Chiglitazar on the Protection Against Sepsis-Induced Acute Lung Injury"

_metabolites, 2025, doi:10.3390/metabo15050290_

Round 1
Reviewer 1 Report
Comments and Suggestions for Authors
- Warburg effect usually refers to aerobic glycolysis seen in cancer cells, but in sepsis, it's about immune cell metabolic reprogramming (macrophages). Here, it's creating confusion.
- The author have listed sphingomyelins, LPCs, ceramides etc., but it’s not explained how each contributes differently (pro-inflammatory vs anti-inflammatory lipids). Also, specialized pro-resolving mediators (SPMs) like resolvins are not mentioned.
- There’s some repetition in how groups are described especially Sham, CLP, CLP+Chi... the author could format batch designs as a bulleted list or in a table for clarity.
- Figure 2C shows that the fatal events... between 6 to 24 h → Should be 6 and 24 h (use and for time ranges).
- CLP+Pio group... showed alveolar wall thickening and mild inflammatory cell infiltration. This reads contradictory unless the author clarify it's less severe than CLP.
- Clarify Chi 10 means 10 mg/kg chiglitazar early on for reader clarity.
- Add n = 3 clearly when referencing sample sizes in figure legend or main text.
- Mitochondrial dysfunction is a central feature of metabolic collapse in sepsis (ROS, ETC inhibition, loss of membrane potential), but it's missing or lightly touched.
- Claiming that the lung is the "primary target organ" in all cases of sepsis is an overstatement. In reality, multiple organ systems may be involved depending on the cause and host response kidneys, liver, heart, and brain also frequently fail.
Reviewer 2 Report
Comments and Suggestions for Authors
This paper demonstrates that chiglitazar alleviates sepsis-associated acute lung injury (SALI) by restoring NAD+ and triglyceride (TG) synthesis through the SIRT1/PGC-1α/PPARα/GPAT3 axis, highlighting the potential of balancing metabolism as a promising therapeutic strategy for managing SALI. The authors provide a comprehensive insight into the metabolic changes during sepsis and the effects of chiglitazar intervention by combining metabolomic, lipidomic analyses, and biological experiments. The following suggestions are proposed:
Major Comments:
-
The authors identified the beneficial effects of chiglitazar (a pan-PPAR agonist) against sepsis-induced acute lung injury (SALI) and used pioglitazone (a PPAR-γ-specific agonist) as the positive control (Line 349). However, their results emphasize chiglitazar's effects through the SIRT1/PGC-1α/PPARα/GPAT3 axis, with PPARα being the central mediator. Is it appropriate to use a PPAR-γ-selective agonist (pioglitazone) as the positive control in this experimental design?
-
Lines 450-460: "Due to the fact that SIRT1 is the enzyme converting NAD+ into nicotinamide (NAM), the elevated NAD+ levels in chiglitazar-pretreated CLP rats were thought to… Such findings suggest that the activity of SIRT1 was not affected by chiglitazar's actions." This section is unclear and requires explicit clarification.
-
The authors reported that SIRT1 was upregulated in the CLP+Chi group compared to the CLP group (Fig. 6), and NAD+ levels were also elevated (Fig. 4). However, since SIRT1 is the enzyme responsible for converting NAD+ into NAM, the observed increase in NAD+ without a corresponding rise in NAM is conceptually inconsistent. How can this be explained? Additionally, the introduction of the SIRT1/PGC-1α/PPARα/GPAT3 axis appears mechanistically disjointed.
Minor Comments:
-
The introduction section of the Abstract does not sufficiently establish the significance of studying SALI or clarify the role of metabolic dysregulation in SALI pathogenesis.
-
In the methodology section of the Abstract, please clarify whether lung tissue, serum, or other samples were analyzed via GC-MS/LC-MS.
-
Figure 4A (PCA plot) requires enhanced statistical rigor. Please add 95% confidence ellipses to visualize inter-group variability. Additionally, perform PERMANOVA/ADONIS testing to quantify the statistical significance of group separations, reporting F-values and p-values in the figure legend.
Reviewer 3 Report
Comments and Suggestions for Authors
The authors made good contribution regarding submitted manuscript. Authors have strong knowledge about the hot topic of the medicinal chemistry. The pictorial quality is remarkable. The contents of the paper are reliable and support to the results with health discussion.
Reviewer 4 Report
Comments and Suggestions for Authors
The article of Lu L.L. et al represents an excellent example of new potential applications of chiglitazar for protection against sepsis induced acute lung injury (SALI) based on a comprehensive analysis of its mechanism of action on the cell, based on the use of metabolomics and lipidomics integrated analysis by LC-mass spectrometry and GC- mass spectrometry methods. Chiglitazar, a well-known drug was widely used for glycemic control in diabetes treatment but its application for preventing mortality of rats in SALI through balancing sepsis induced metabolic disorders was shown for the first time. SALI state was induced by CLP surgery and compared with a SHAM state. Treatment with pioglitazone was used as a positive control. The authors show the protective effect of chiglitazar on life span, on the state of lung tissue revealed by pathohystological and immunohistochemical analysis, on the levels of pro-inflammatory cytokins IL-1β, IL-6, TNF-α quantified by the authors in BALF which decreased in a dose-dependent manner, q-PCR and Western blotting indicated resuscitation of NAD+ levels, stabilization of L-Kynurenine and Quinolinate metabolism, reduction of excessive fatty acids that uncouple mitochondria, and increase of TGs. Application of Ex-527, SIRT1 inhibitor, to normal human lung cells (BEAS-2B) and one group of rats resulted in reversing the protective effects of chiglitazar which allowed the authors to elucidate the mechanism of chiglitazar action supposedly associated with SIRT1/PGC-1 α/PPARα/GPAT3 signaling axis defining normal cross-talk of mitochondria with the cell. An additional advantage of the work is the fact that the comprehensive analysis of cellular metabolism carried out by the authors allows us to look for the first time into the molecular biological processes caused by the development of sepsis.
Also a good impression the work makes in data presentation. In Fig. 1 and 2B, the authors presented visual diagrams of the algorithm for dividing animals into groups and the general plan for setting up experiments. All the conclusions are supported by the results obtained. I would especially like to note the good English language of the article. The only technical slips are in Fig.1A box 1 and 5 Obsorvation should be changed for Observation.
The article is interesting, contains new information and suits well for publication in Metabolites
Round 2
Reviewer 1 Report
Comments and Suggestions for Authors
The authors addressed all the questions and incorporated the suggestions in the manuscript. The manuscript may be accepted for publication in the present form.